# *Bifidobacterium animalis* subsp. *lactis* Probio-M8 alleviates abnormal behavior and regulates gut microbiota in a mouse model suffering from autism

Zhuangzhuang Miao,[1,2] Lin Chen,[3] Yong Zhang,[4] Jiachao Zhang,[3] Heping Zhang[1,2]

**ABSTRACT** Probiotics can effectively improve a variety of neurological diseases, but there is little research on autism, and the specific mechanism is unclear. In this study, shotgun metagenomics analysis was used to investigate the preventive and therapeutic effects of *Bifidobacterium animalis* subsp. *lactis* Probio-M8 on autism. The results showed that Probio-M8 treatment significantly alleviated valproate (VPA)-induced autism in mice, with autistic symptoms characterized by increased stereotyped behaviors such as grooming, reduced learning ability, and decreased desire to socialize. Further studies have found that Probio-M8 can alleviate autism by optimizing gut microbiota diversity and regulating metabolic levels. Probio-M8 regulates gut microbiota structure by increasing the abundance of beneficial bacteria such as *Bifidobacterium globosum* and *Akkermansia muciniphila*. In addition, Probio-M8 regulates metabolic activity by increasing levels of choline, which corrects CAZy disorders. In conclusion, Probio-M8 is therapeutic in the VPA-induced autism mouse model by regulating the gut microbiome and metabolic levels.

**IMPORTANCE** Individuals with autism often exhibit symptoms of social invariance, obsessive-compulsive tendencies, and repetitive behaviors. However, early intervention and treatment can be effective in improving social skills and mitigating autism symptoms, including behaviors related to irritability. Although taking medication for autism may lead to side effects such as weight gain, probiotics can be an ideal intervention for alleviating these symptoms. In this study, we investigated the effects of Probio-M8 intervention on the behavior of autistic mice using an open-field test, a three-chamber sociability test, and a novel object recognition test. Metagenomic analysis revealed differences in gut microbiota diversity among groups, predicted changes in metabolite levels, and functionally annotated CAZy. Additionally, we analyzed serum neurotransmitter levels and found that probiotics were beneficial in mitigating neurotransmitter imbalances in mice with autism.

**KEYWORDS** probiotics, autism, gut microbiota, *Bifidobacterium animalis* subsp. *lactis* Probio-M8

Autism is an early-onset neurodevelopmental disorder that seriously affects the lives of its sufferers. According to statistics, the prevalence of autism spectrum disorder (ASD) in children has been increasing annually since 2018 (1), causing widespread medical and social problems. The prevalence of autism in children and adolescents is high, and the co-morbid symptoms (2) mainly include anxiety, depression (3), repetitive stereotypic behaviors (4), substance use disorder (5), etc. In addition, ASD is regularly accompanied by symptoms of gastrointestinal abnormalities characterized by gut infections, increased gut permeability, and altered microbiota composition (6).

Address correspondence to Heping Zhang, hepingdd@vip.sina.com.

Zhuangzhuang Miao and Lin Chen contributed equally to this article. Author order was determined on the basis of seniority.

The authors declare no conflict of interest.

See the funding table on p. 14.

Fortunately, autistic patients can be treated with medications such as antidepressants, stimulants, and behavioral therapy to improve autism symptoms (7). When patients with autism show agitation or aggressive behavior, they need to follow medical advice to take drugs such as risperidone and aripiprazole for treatment (8, 9). However, the administration of risperidone medication can easily cause adverse effects such as anxiety, drowsiness, and constipation, while the administration of aripiprazole can produce adverse effects in the digestive and cardiovascular systems (10, 11).

Compared to drugs, probiotics not only have a few side effects when preventing and treating diseases but can also promote the healing of various diseases (12, 13). Probiotics are living microbiota that can have a beneficial effect on the body when consumed in certain amounts (14). Some studies have shown that probiotics can improve gastrointestinal symptoms, bringing a palliative effect on some psychiatric disorders (15–17). The signals produced by the microbiota are closely related to neurodevelopment and social behavior. The gut microbiota has been shown to modulate host behavioral functions and exhibit key roles in central nervous system-related behaviors (18, 19). In addition, gut microbiota in ASD patients can have an impact on the early neurodevelopment of pediatric patients through the gut-brain axis, thus affecting their autistic behavior (20). In recent years, the intervention of the human gut microbiota through the intake of probiotics to avoid the side effects of conventional drugs to treat psychiatric disorders has become a hot research topic, and successful clinical trials have been reported in patients with ASD (21–23).

*Bifidobacterium* is a safe probiotic widely used in dairy products to reduce harmful bacteria and increase beneficial bacteria, which can play a role in treating certain diseases (24). *Bifidobacterium animalis* subsp. *lactis* Probio-M8 (Probio-M8) was obtained from colostrum samples of healthy lactating women in Inner Mongolia who showed good tolerance to gut juice, gastric acid, and bile salts of the digestive system (25). The combined administration of Probio-M8 has been reported to enhance the therapeutic effect on Parkinson's disease (PD), relieve anxiety, and reduce gastrointestinal symptoms (26). In addition, it has been shown that Probio-M8 alleviates cognitive impairment and prevents dysbiosis of the gut microbiota in APP/PS1 mice (27). Autistic people have more gastrointestinal disorders and anxiety symptoms than the general population (28), and Probio-M8 may have the potential to solve this problem.

Therefore, Probio-M8 may be a good option for preventing or treating autism. In this study, we used Probio-M8 to intervene in a mouse model of ASD and examined changes in gut microbiology, behavioral performance, and neurotransmitters in mice after the intervention. The results of this study may provide new insights into how probiotics may intervene in the treatment of patients with ASD.

## RESULTS

### *Bifidobacterium animalis* subsp. *lactis* Probio-M8 alleviates unusual behavior in a mouse model of ASD

The ASD mouse model in this study was obtained using valproate (VPA) intervention in pregnant females. The experimental grouping is shown in Fig. 1a. At 3 weeks old, the mice in the model group spent significantly less time in the central region than those in the control ratio, while the total distance of movement and **the frequency of grooming** increased significantly. During the same period, **the frequency of grooming** was strongly significant ($P < 0.01$), and the total distance of movement was extremely significant ($P < 0.001$) in the prevent group compared to the model group (Fig. 1b). At 9 weeks old, mice in the model group spent significantly less time in the central region than those in the control group. The total distance of locomotion and the frequency of grooming were significantly reduced in the treatment and prevent groups compared to the model group ($P < 0.05$) (Fig. 1c). In the open field test, at 3 and 9 weeks, the mice in the model group showed significant ASD-related behavior. In contrast, the frequency of grooming sessions and the total distance traveled by the prevented and treated mice had been significantly reduced ($P < 0.05$), suggesting a reduction in their ASD-related behavior.

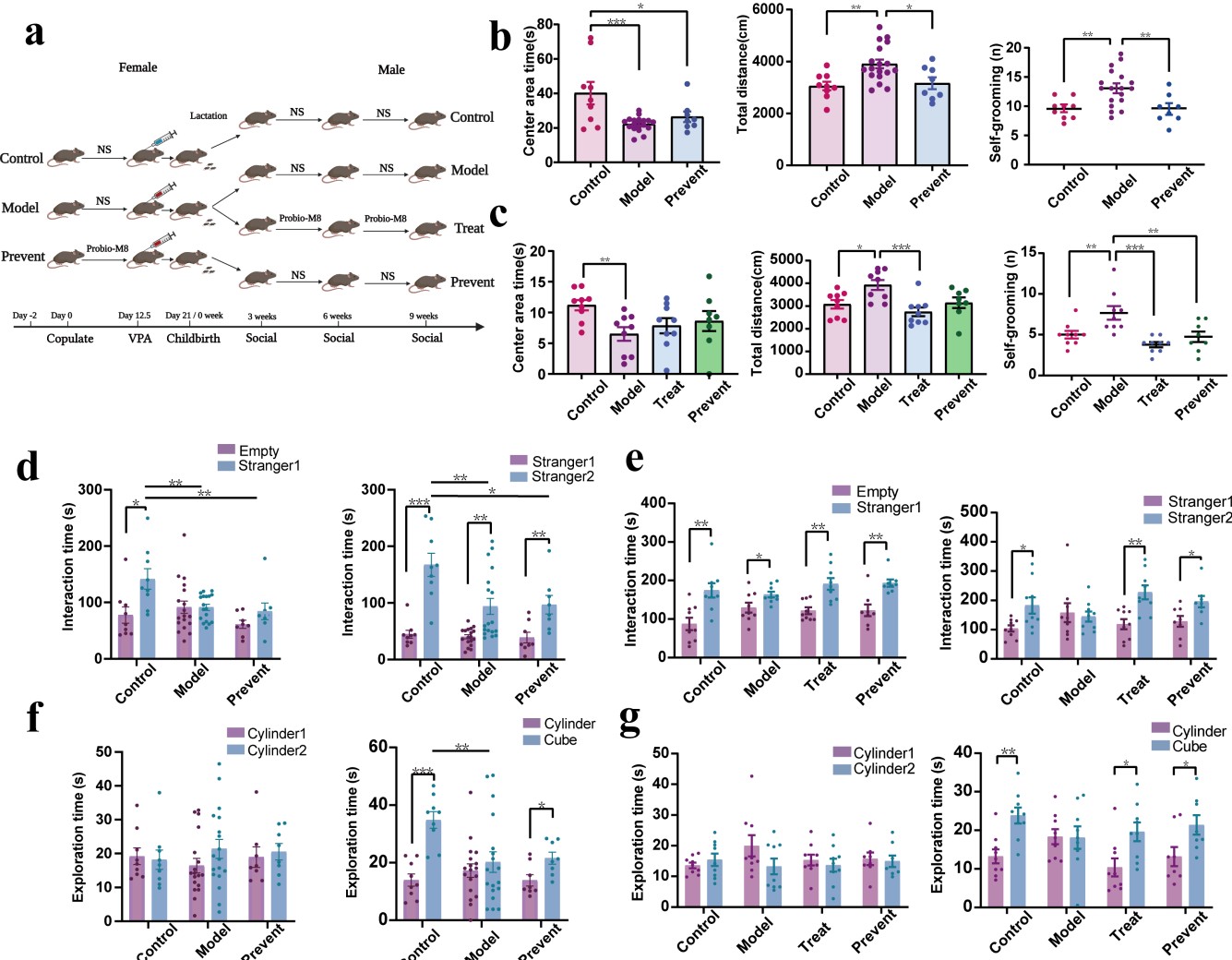

**FIG 1** Experimental grouping and results of the mouse behavior experiment. (a) Animal experimental design: procedure for intervention in pregnant mice and their litters. (b) Results of an open field test in 3-week-old mice. (c) Results of an open field test in 9-week-old mice. (d) Results of the three-chamber sociability test in 3-week-old mice. (e) Results of the three-chamber sociability test in 9-week-old mice. (f) Results of the novel object recognition test in 3-week-old mice. (g) Results of the novel object recognition test in 9-week-old mice. A one-way analysis of variance followed by Least Significant Difference post hoc analysis was used for multiple pairwise comparison. *, $p < 0.05$; **, $p < 0.01$; and ***, $p < 0.001$. Data are plotted as ±standard error of the mean. NS means normal saline.

A three-chamber sociability test can be used to evaluate the social competence of mice. For 3-week-old mice, the model group was significantly less exposed to new mice than the control group, which can be seen as poor socialization (Fig. 1d). For 9-week-old mice, no significant differences were found in the time spent exploring old and new mice in the model group. In contrast, the control, treatment, and prevent groups had significantly increased exposure to new mice and had good socialization skills ($P < 0.05$) (Fig. 1e). Through a three-chamber sociability test, we found that both preventively and therapeutically treated autistic mice possessed better socialization skills, suggesting that the administration of Probio-M8 improves socialization in autistic mice.

Novel object recognition tests can be used to evaluate how well animals learn and remember. There is no significant difference in the model group's exploration time for new and old objects at 3 weeks old. Meanwhile, the control group and prevent group were more inclined to explore new objects ($P < 0.05$) (Fig. 1f). At 9 weeks old, we found no significant difference in the time spent exploring new and old objects in the model group, while the control, treatment, and prevent groups all had significantly more

exposure time to new objects and better learning memory ($P < 0.05$) (Fig. 1g). Experiments showed that mice with autism receiving preventive and therapeutic interventions were more inclined to be exposed to new objects, suggesting improved learning and memory skills.

In conclusion, the 3-week-old model mice showed stereotyped behaviors similar to autism, with poor social skills and learning memory, indicating that the model was successfully constructed at this time. The prevent group showed initial relief in some autism symptoms and learning and memory abilities. At 9 weeks old, the model group still showed autism symptoms and poor social, learning, and memory abilities, while the prevent group and the treat group both had significant remission of autism symptoms, especially the treat group. Probio-M8 was found to be helpful in the early prevention and later treatment of VPA-induced autism in mice and was able to correct autistic-like behavior in mice.

## Intake of *Bifidobacterium animalis* subsp. *lactis* Probio-M8 probiotics optimizes gut microbiota structure in autistic mice

Fecal samples were collected from 3-week-old, 6-week-old, and 9-week-old mice (control, model, preventive, and treat groups) for metagenomic analysis. As shown in Fig. 2a, the Shannon index of the prevent group was significantly elevated compared to the

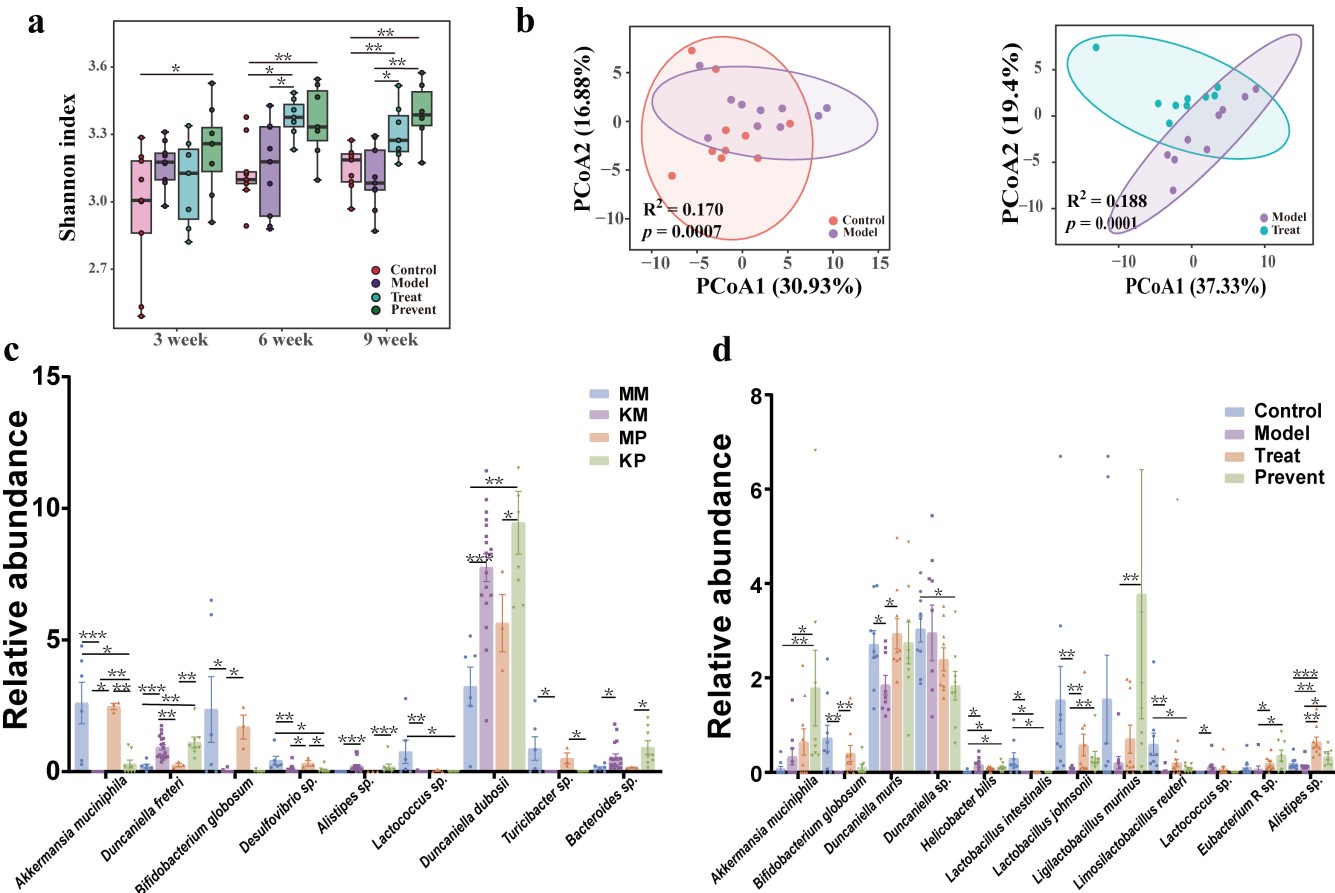

FIG 2 Effect of Probio-M8 on the gut microbiota of mice. (a) Shannon index for each group of mice throughout the experimental cycle. Data are plotted as ±standard deviation. (b) Species-level PCoA based on weighted UniFrac distances with the Adonis test. (c) The abundance of gut microbiota at the species level in mothers and 3-week-old littermates. MM stands for mothers in the model group; KM stands for littermates in the model group; MP stands for mothers in the prevention group; and KP stands for littermates in the prevention group. Data are plotted as ±standard error of the mean. (d) The abundance of gut microbiota at the species level between groups of 9-week-old littermates. Data are plotted as ±standard error of the mean. The Wilcoxon rank-sum test with Bonferroni correction was used for multiple pairwise comparisons. *, $p < 0.05$; **, $p < 0.01$; and ***, $p < 0.001$.

control group at 3 and 6 weeks and significantly elevated compared to both the control and model groups at 9 weeks ($P < 0.05$). The treat group was significantly elevated compared to the control and model groups at 6 and 9 weeks ($P < 0.05$). As shown in Fig. 2b, principal coordinate analysis (PCoA) based on Bray-Curtis distance showed that the structure of the gut microbiota in model mice was significantly different from that of control and treated mice. By analyzing the diversity of the gut microbiota in mice, we found that mice treated with probiotics were able to alter the gut microbiota composition of mice with autism.

Metagenomic species analysis of fecal samples from mothers and 3-week-old litters showed that the beneficial bacteria *Lactococcus* decreased and the harmful bacteria *Duncaniella dubosii* increased in the model group compared with the mothers ($P < 0.05$). Compared with the KM group, the beneficial bacteria *Akkermansia muciniphila* and *Bacteroides* sp. increased in the KP group ($P < 0.05$) (Fig. 2c). The metagenome of fecal stool in 9 weeks was compared. Compared with the model group, the abundance of *Lactobacillus johnsonii*, *Eubacterium R* sp., and *Akkermansia muciniphila* in the prevent group increased significantly. Furthermore, in the treat group, the beneficial bacteria *Bifidobacterium globosum* and *Lactobacillus johnsonii* increased significantly ($P < 0.05$) (Fig. 2d). In conclusion, early intervention and late treatment with Probio-M8 affected the gut microbiota composition profile and optimized species composition in ASD mice.

## Intake of Probio-M8 corrects gut microbiota metabolite imbalances in autistic mice

Metagenomic sequences of 9-week-old mice were used to predict the gut microbiota metabolic profile. By metabolic prediction, we found that the contents of cholesterol and X3 methylxanthine in the prevent group were significantly increased compared with the control group (Fig. 3). Similarly, the contents of ketodeoxycholate in the treat group were significantly increased compared with the control group ($P < 0.05$). The levels of lithophanate, nicotinate, and propionate were elevated in the prevention group compared to the model group and were significant ($P < 0.05$). As shown in Fig. 3c, the imidazole propionate in the treat group was significantly increased. Abnormal cholesterol levels can lead to various neurological diseases, and low cholesterol often leads to ASDs. Nicotinate can improve mental symptoms such as memory loss. In terms of metagenomic prediction, this phenomenon implies that probiotic administration may affect metabolite levels in autistic mice. Further research is needed to determine the exact mechanism.

## Intake of Probio-M8 corrects CAZy disruption in autistic mice

CAZy degrades carbohydrates, and it is a non-negligible presence for both the gut microbiota and the host. Therefore, we used the CAZy database to annotate metagenomics functionally to explore potential roles. As shown in Fig. 4a, the content of auxiliary activities (AAs) was significantly higher in both the prevent and control groups than in the model group; the content of glycosyl transferases (GTs) was significantly higher in the control group than in the model group. This suggests that CAZy is disrupted in mice suffering from autism. In contrast, gut CAZy levels were elevated in all of the mice treated with probiotics, implying that probiotic use may be able to restore the disruption of CAZy caused by autism.

As shown in Fig. 4b, compared with the control group, the autistic mice down-regulated 26 genes and up-regulated 4 genes. Compared with the model group, the treat group up-regulated 25 genes and down-regulated 7 genes (Fig. 4c). The prevent group of mice up-regulated 12 genes and down-regulated 11 genes (Fig. 4d). Genes down-regulated in autistic mice such as CBM25, CBM66, CBM77, CE5, GH128, GH13_32, GH30_5, GH43_24, GT14, PL11, PL12, PL26, and PL4. We found that most of these genes belong to the CBMs (carbohydrate-binding modules), GHs (glycoside hydrolases), and PLs (polysaccharide lyases), all of which were up-regulated in therapeutic or preventive

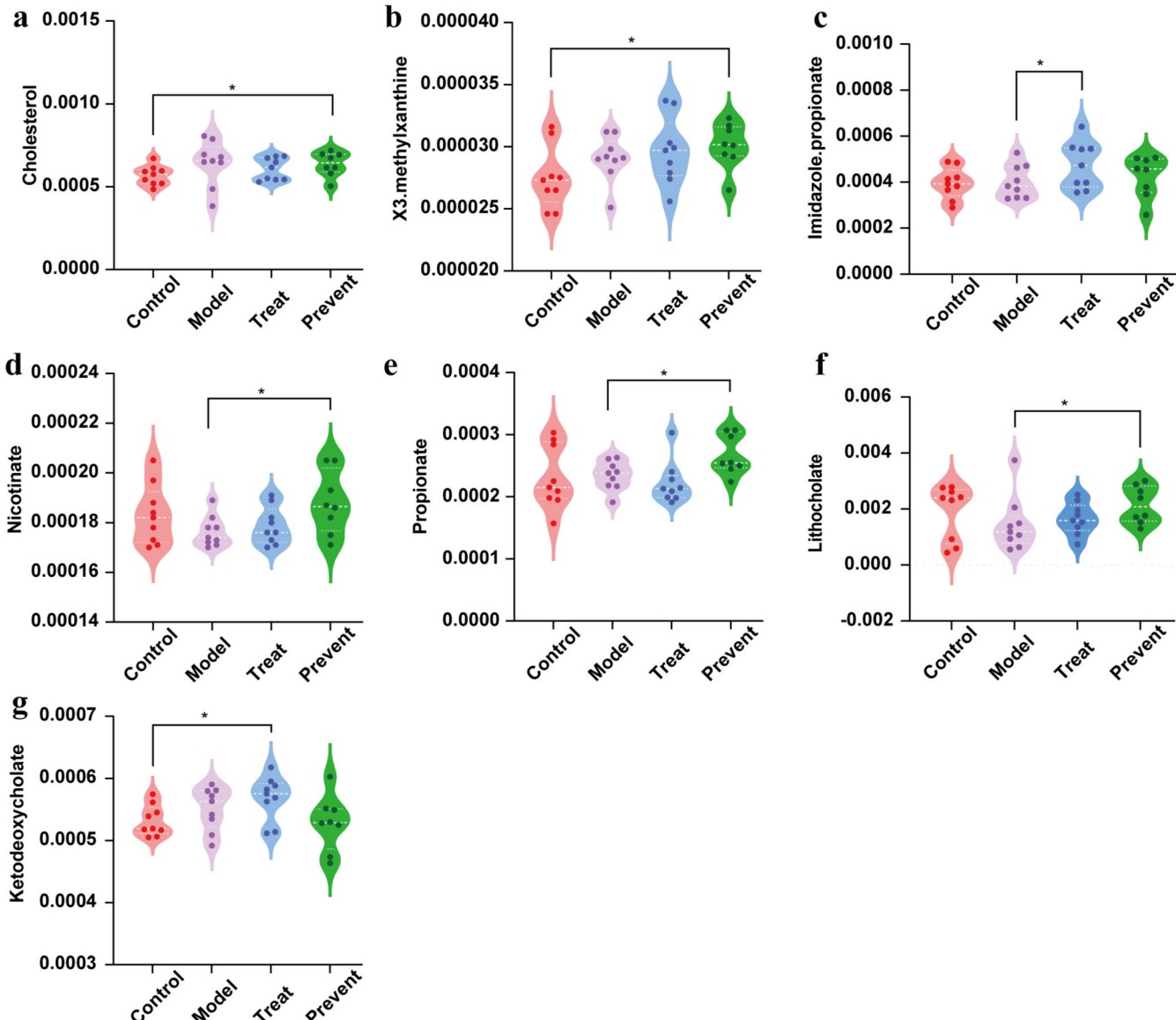

**FIG 3** Prediction of metabolite compound abundance in 9-week-old mice. (a) The level of cholesterol. (b) The level of X3.methylxanthine. (c) The level of imidazole propionate. (d) The level of nicotinate. (e) The level of propionate. (f) The level of lithocholate. (g) The level of ketodeoxycholate. Dots indicate single mice (Wilcoxon rank-sum test). *, $p < 0.05$. Data are plotted as ±standard deviation.

mice. This suggests that taking probiotics can restore CAZy and its dysfunction due to autism.

## Effect of Probio-M8 intake on neurotransmitters in autistic mice

By using the serum for neurotransmitter assays, we found significantly lower levels of serotonin, its precursor tryptophan, and its end product 5-hydroxyindoleacetic acid in the serum of the model group than in the serum of the control group ($P < 0.05$) (Fig. 5a through c). Serotonin levels were significantly higher in the prevent and treat groups than in the model group. Glutamate and 3-hydroxytyramine (dopamine) levels were significantly increased in the control group compared to the model group. At the same time, glutamate levels were also increased in the prevent group and treat group compared to the model group. Choline levels were observed to be elevated in the treat group and prevent groups compared to the model group ($P < 0.01$) (Fig. 5f). The above situation indicates that the metabolism of serotonin in ASD mice is disturbed, which can

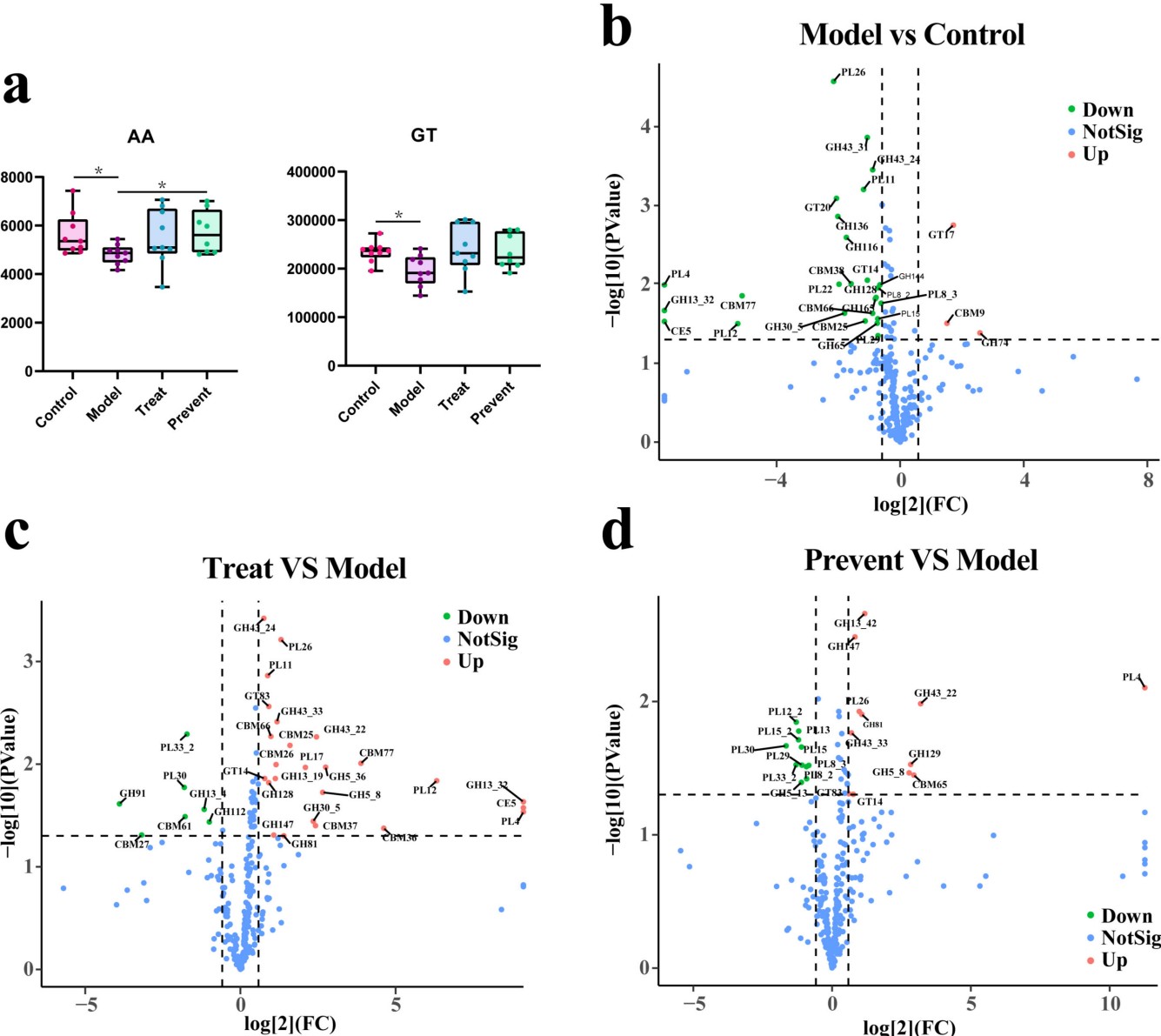

FIG 4 CAZy functional annotation results. (a) Gene content of AA and GT between groups. (b) Volcano plots of the model group versus the control group. (c) Volcano map of the treat group versus the model group. (d) Volcano map of the prevent group versus the model group. *, $p < 0.05$.

be alleviated by Probio-M8 to prevent and treat ASD. In addition, the imbalance of glutamate, dopamine, and choline appeared in ASD mice, and Probio-M8 can prevent, treat, and effectively relieve the symptoms of dopamine and choline imbalance.

## Correlation network

We evaluated the relationship between gut microbiota and metabolites at 3 and 9 weeks (Fig. 6a and b). At 3 weeks, *Akkermansia muciniphila* showed a significant negative correlation with imidazole propionate and a significant positive correlation with nicotinate; *Duncaniella muris* showed a significant negative correlation with X3 methyl-xanthine. At 9 weeks, *Akkermansia muciniphila* showed a significant negative correlation with imidazole propionate, and *Ligilactobacillus murinus* showed a significant positive correlation with nicotinate. Nicotinate can improve mental symptoms such as memory loss (29). Imidazole propionate production is detrimental to insulin signaling, leading to type II diabetes, and disrupts the brain's cognitive functions of learning (30, 31). The

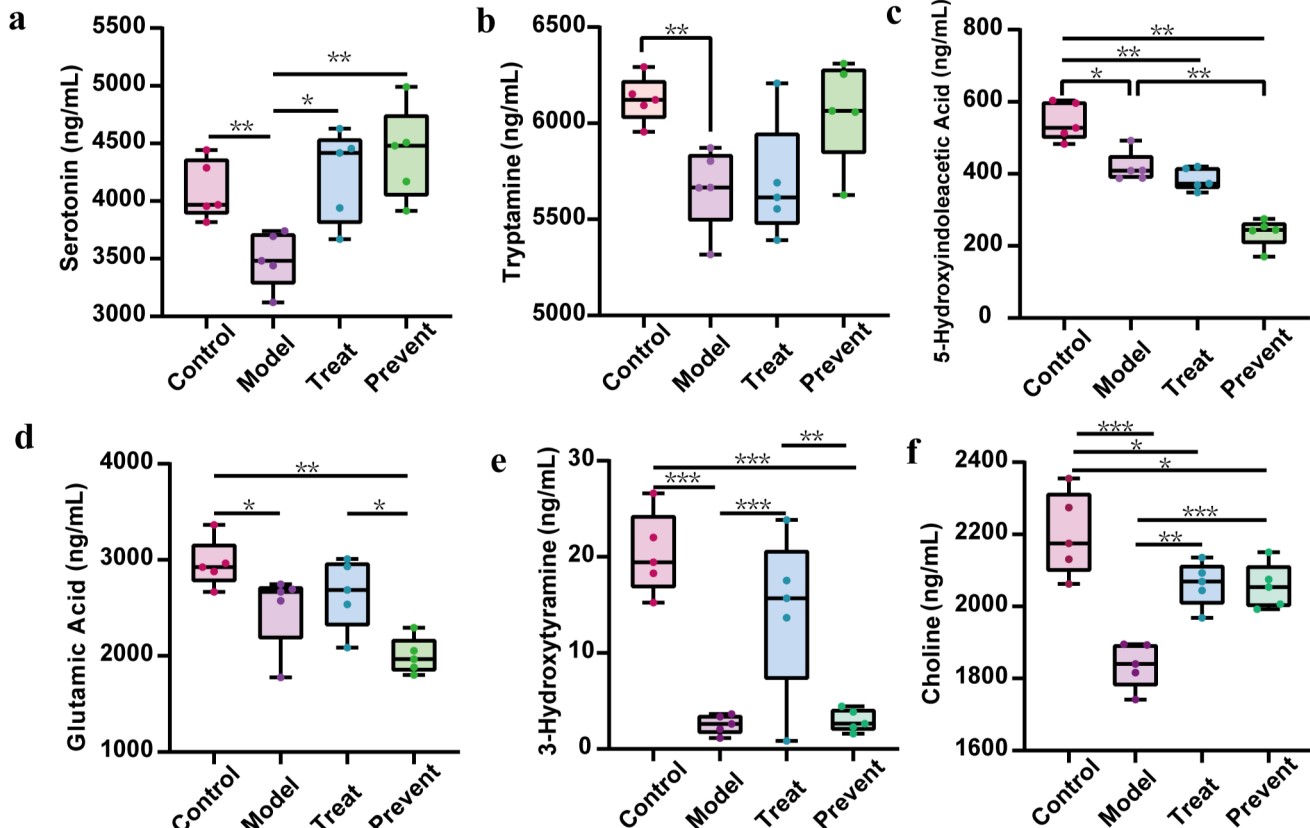

**FIG 5** Serum neurotransmitter abundance in mice aged 9 weeks (Wilcoxon rank-sum test). (a) The level of serotonin. (b) The level of tryptamine. (c) The level of 5-Hydroxyindoleacetic acid. (d) The level of glutamic acid. (e)The level of 3-Hydroxytyramine. (f) The level of choline. Dots indicate single mice. *, $p < 0.05$; **, $p < 0.01$; and ***, $p < 0.001$. Data are plotted as ±standard deviation.

results suggest that probiotics may help to manage the gut microbiota and metabolites to correct symptoms in mice with autism.

For a more in-depth study, we analyzed the correlation between gut microbiota species, metabolites, and behavioral experiments in the 9-week prevent group of mice. It was shown in Fig. 6c that the residence time of mice in the center was positively correlated with beneficial bacteria *Bifidobacterium globosum* and *Duncaniella*, and it was negatively correlated with harmful bacteria *Helicobacter bilis* ($r < -0.3$). Total exercise distance and beneficial bacteria *Duncaniella muris*, *Eubacterium R* sp., and *Lactobacillus johnsonii* were negatively correlated. Grooming behavior was negatively correlated with *Bifidobacterium globosum* ($r < -0.3$). Social behavior was positively correlated with *Duncaniella muris*, propionate, and X3 methylxanthine. The beneficial bacteria *Akkermansia muciniphila* was positively correlated with the X3 methylxanthine and ketodeoxycholate ($r > 0.3$). We hypothesize that supplementation with Probio-M8 corrects autistic behaviors such as frequent grooming and poor social skills by regulating the composition of the gut microbiota and restoring the balance of autistic metabolites such as X3 methylxanthine.

Subsequently, we analyzed the correlation between gut microbiota species, metabolites, and behavioral experiments in the 9-week treatment group of mice. Correlation analysis of metagenomic, metabolite, and behavioral experiments in the treat group of mice was performed. As shown in Fig. 6e, the center residence time of mice was positively correlated with beneficial bacteria *Bifidobacterium globosum* ($r > 0.3$) and *Duncaniella*, and it was negatively correlated with harmful bacteria *Helicobacter bilis* ($r < -0.3$). Total exercise distance was negatively correlated with the beneficial bacteria *Duncaniella muris*, *Eubacterium R sp.*, *Bifidobacterium globosum*, and *Lactobacillus johnsonii*. Grooming

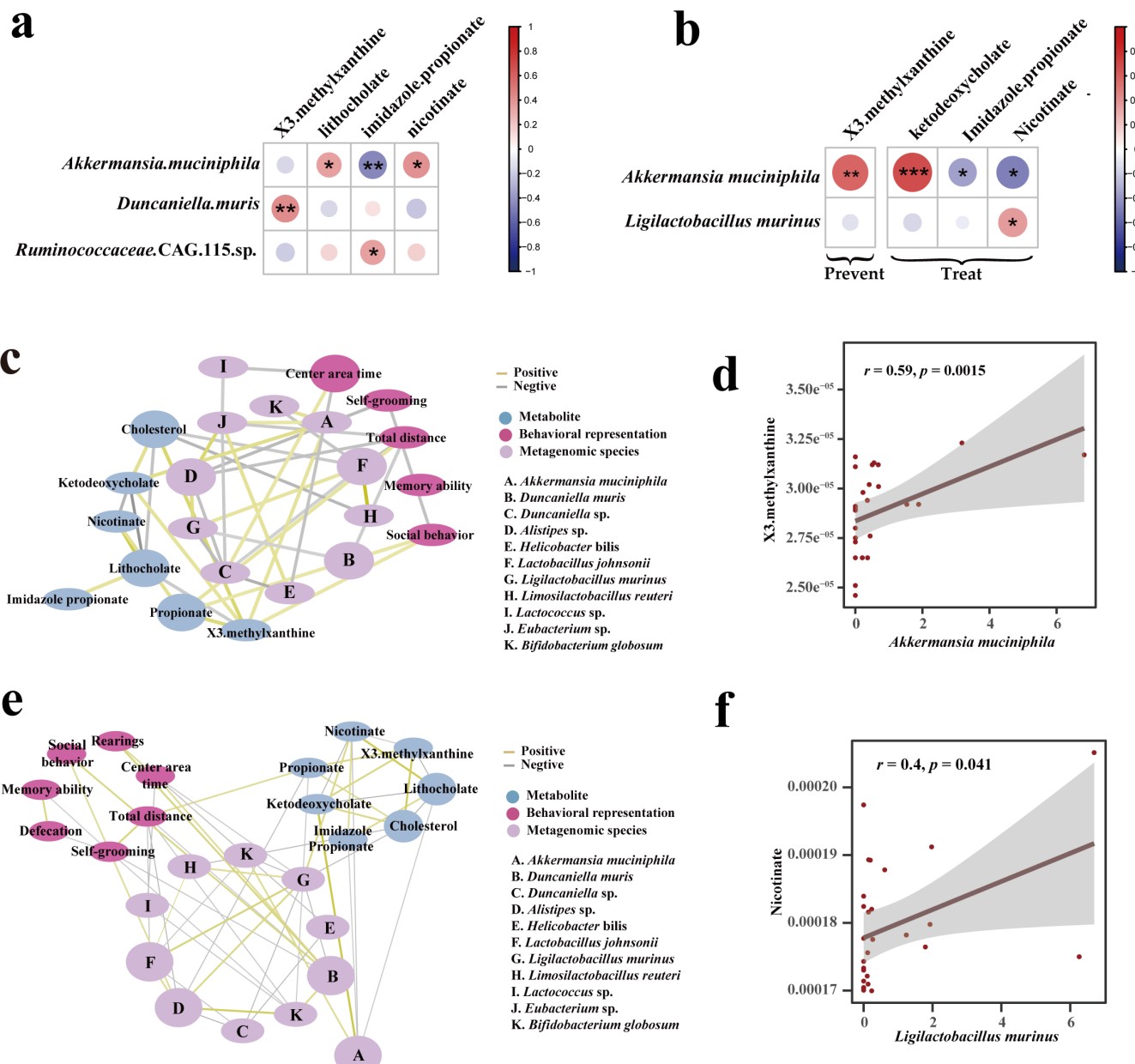

**FIG 6** Correlation analysis. (a) Correlation heatmap of metabolite and gut microbiota in 3-week mice (red: positive correlation; blue: negative correlation). (b) Correlation heatmap of metabolites and gut microbiota in 9-week mice. (c) Correlation network diagram of metabolites, behavioral representation, and gut microbiota in 9-week prevent group mice. (d) Regression plot of key metabolite and gut microbiota correlations in mice in the 9-week prevent group. (e) Correlation network diagram of metabolites, behavioral representation, and gut microbiota in 9-week treat group mice. (f) Regression plots of key metabolite and gut microbiota correlations in mice in the 9-week treat group. In the correlation analysis, all are plotted based on the Spearman index. Where the width of the network diagram connecting lines is proportional to the strength of the correlation (yellow: positive; gray: negative).

behavior was negatively correlated with *Eubacterium R sp.* ($r < -0.3$). Social behavior was positively correlated with *Duncaniella muris* ($\underline{r} > 0.3$). The beneficial bacterium *Limosilactobacillus reuteri* was positively correlated with *Lactobacillus johnsonii* and *Ligilactobacillus murinus*. Nicotinate was positively correlated with *Ligilactobacillus murinus* ($r > 0.3$) (Fig. 6f). We hypothesized that Probio-M8 supplementation could influence behavioral symptoms and metabolic levels in autism by modulating the gut microbiota. Overall, there was a decrease in potentially harmful bacteria and an increase in potentially beneficial bacteria. This may suggest that treating autism with Probio-M8 may optimize the structure of the gut microbiota by altering the abundance of both practical and

harmful bacteria, thereby reducing stereotypic behaviors associated with autism and increasing the desire to socialize. In addition, altering the gut microbiota changes metabolites such as nicotinate levels, which in turn influences autism-related behaviors.

## DISCUSSION

The experimental mice in this study showed increased stereotypic behavior, increased total distance of movement, a tendency to be in the center area, and poor social learning ability in response to autism induced by using VPA, conditions that indicate the successful establishment of a mouse model of autism. Compared with the model group, both the prevent group and treat group exhibited good social learning behavior, reduced stereotypic behavior, optimized gut microbiota structure, and increased beneficial neurotransmitters and metabolites. In summary, our findings suggested that after 9 weeks, prevention or treatment with Probio-M8 somewhat alleviated autism and its associated behaviors.

After the intervention of VPA in pregnant female mice, the mice showed different degrees of autism at 3 and 9 weeks. These mice showed more aversion to staying in the central area of the open field experiment, more exercise, and more stereotypical behaviors such as grooming and poor social, learning, and memory abilities. In contrast, the total distance exercised and the frequency of grooming sessions decreased after the intervention with Probio-M8, and social and learning memory skills improved. Numerous studies have pointed out that supplementation with appropriate amounts of probiotics can be effective in improving the behavioral symptoms of depression, Alzheimer's disease (AD), ASD, and other psychiatric disorders (32–34). Similarly, this study found that Probio-M8 can relieve autistic behavior symptoms of ASD and improve social ability, learning, and memory ability in male mice to varying degrees.

The gut microbiota in patients with different neurological disorders is very different. Several studies have reported that in a metagenomic analysis of patients with major depression, bipolar disorder, and Parkinson's disease, *Bifidobacterium* was found to be significantly increased in the patients (35–37). However, in the gut of individuals with autism, several studies have reported evidence of significantly lower levels of *Bifidobacterium* (38–40). This phenomenon is consistent with our research, where we found that *Bifidobacterium globosum* was significantly elevated in mice using probiotics at 9 weeks. Nevertheless, further studies are needed to verify the mechanism of the intervention of *Bifidobacterium* effects on ASD. At 3 and 9 weeks old, we found a significant increase in the abundance of *A. muciniphila* in the intestine of mice prophylactic with Probio-M8 compared to mice with ASD. There is some support in the literature that *A. muciniphila* and *Bifidobacterium* are found at lower levels in the gut of individuals with autism than in the general population (41, 42). *A. muciniphila* helps maintain the gut barrier and alleviate gut inflammation, as well as helping to regulate metabolites to modulate the central nervous system (40, 43, 44). We hypothesized that supplementation with Probio-M8 could increase the abundance of *A. muciniphila* in autistic patients to alleviate ASD symptoms. This suggests that probiotics play a pleiotropic role in alleviating autism symptoms by regulating the gut microbiota through a network matrix. However, in-depth studies are needed to confirm the mechanism of its effect.

Abnormal cholesterol levels are strongly associated with the development of many neurological disorders. Benachenhou et al. found that a low cholesterol level is associated with ASD (45). In our study, the prevent group mice maintained high cholesterol levels. Other studies have shown that nicotinate can be used to treat autism and related symptoms, which has been in the stage of clinical trials (46). The exact mechanism needs to be determined by further actual measurements. Similarly, we found in our experiments that nicotinate was significantly elevated in mice taking Probio-M8. In addition, CAZy functional annotation results showed that CHs and CBMs were up-regulated in mice treated with Probio-M8. GHs supplementation produces glycan chains, and the mucus layer of mucin glycans protects the epithelium from microorganisms and luminal compounds, thus providing a defensive effect, which is essential for maintaining

gut barrier functions (47, 48). CBMs can increase catabolic substrate efficiency by attacking difficult-to-access substrates. In summary, probiotics help restore gut CAZy's disturbance and maintain gut microbiota homeostasis in autistic mice (49, 50). This suggests that probiotics help restore gut CAZy disturbances and maintain gut microbiota homeostasis in autistic mice.

Typically, there is a neurotransmitter imbalance in ASD mice. The ASD mouse model in this study showed signs of dopamine and choline imbalance, and these levels were significantly reduced compared with control mice. Supplementation with Probio-M8 significantly increased dopamine and choline levels in mice in both the prevention and treatment groups compared to the model group. Jennings and Basiri were consistent with this study's results on reducing serum choline content in patients with ASD (51). Gut microbiota in people with autism may affect the levels of metabolites or neurotransmitters, which in turn affect the symptoms of people with autism (52).

Nevertheless, our study still has some limitations. First, our sample size is limited, and more in-depth exploration requires further expansion of the sample size. Second, we only analyzed male mice, and female mice should be included in future studies. Finally, the factors affecting gut microbiota and various metabolites are complex and multiple, and we could not exclude all influencing factors.

Autism and other neurological disorders are potential diseases related to the gut, progressively affecting the immune, metabolic, and nervous systems (53–55). Based on the Pearson correlation coefficient, it is hypothesized that the intake of Probio-M8 may modulate autistic behavioral symptoms and gut microbiota metabolism by altering the abundance of gut microbiota and having beneficial effects on the regulation of neurotransmitters. Therefore, supplementation with Probio-M8 may be a safe and effective treatment for these neurodevelopmental disorders. This study may provide a new perspective on the effects of Probio-M8 on gut microbiota and autistic behavior.

## MATERIALS AND METHODS

### Establishment of a mouse model of ASD

Sixty (20 males and 40 females) 6–8 week-old specific pathogen-free C57BL/6 N mice (Beijing Huafu Kang Biotechnology Co., Ltd.) were prepared for the experiments. According to the previous method, C57BL/6 N mice were adapted to an independent ventilated cage box in the laboratory for a week. The cage was closed at 8:00 in the evening (56). The prepared male mice were mated with the female mice in a cage overnight, and the female mice were observed the following day for the presence of vaginal plugs. If a vaginal plug is observed in the vulva or vagina, the mice are considered mated and possibly pregnant. A total of 40 mice were observed to be pregnant. The methods were performed in accordance with relevant guidelines and regulations and approved by the Special Committee on Scientific Research and Academic Ethics of Inner Mongolia Agricultural University.

The entire experimental design is illustrated in Fig. 1. Pregnant female mice were randomly divided into three groups: the model group ($n = 20$), the control group ($n = 10$), and the prevent group ($n = 10$). As shown in the figure below, the prevent group was given $Bifidobacterium\ animalis$ subsp. $lactis$ Probio-M8 ($2 \times 10^8$ CFU/mL) 2 weeks before modeling, and control and model offspring mice were given saline (Shijiazhuang Sipharmaceutical Co., Ltd.). Since males are more susceptible to autism than females and estrogen is neuroprotective, which would interfere with the experiment, there is a preference for using male littermates for research (57–60). On day 12.5, after the discovery of the vaginal plug, saline was injected intraperitoneally into the control group, whose male mice were selected to serve as the control group in the next stage ($n = 9$). Prevent pregnant mice were injected intraperitoneally with VPA (Sigma, USA) at a 500 mg/kg dose, and after they gave birth, male littermates were selected as the prevent group ($n = 8$). After the female mice in the model group gave birth to their littermates, male littermates were selected and randomly divided into the treat group ($n = 9$) and

the model group ($n$ = 9). Three weeks after birth, mice were continuously instilled with *Bifidobacterium animalis* subsp. *lactis* Probio-M8 ($2 \times 10^8$ CFU/mL) for 6 weeks for the treat group and saline for the control, model, and prevent groups. Fecal samples were collected from each group of littermates at three intervals at 3, 6, and 9 weeks old, and orbital blood collection was performed at 9 weeks. The samples were promptly stored in a −80℃ freezer after collection for subsequent studies.

## Open field test

A digital camera was set up right above the mouse field reaction box (50 × 50 × 45 cm), and its field of view could cover the whole field interior. The mice were positioned in the bottom center area of the box while camera recording and timing were performed, and the movement of the mice was tracked. The camera was stopped after 10 minutes of observation. The inside of the box and the bottom surface were wiped with 75% ethanol before changing animals.

## Three-chamber sociability test

Before the start of the experiment, mice were positioned in the experimental apparatus (60 × 40 × 22 cm) and allowed to acclimatize freely in the three connecting chambers for 10 minutes. A clear glass resin baffle separated the three boxes, and the experimental mice were allowed to move around in the empty chamber for 5 minutes. One mouse (Stranger 1) was randomly placed in a metal cage on one side of the box, and the glass resin baffle between the boxes was removed to allow the experimental mice to have 10 minutes of activity in all the boxes. After that, another unfamiliar mouse (Stranger 2) was placed in an empty metal cage on the other side of the experimental setup. The activity of the experimental mouse was again observed and recorded for 10 minutes. The entire experiment was recorded by a video camera placed above the head.

## Novel object recognition test

The animals were first placed in the laboratory for 1 h to acclimate to the surrounding environment for the formal experiments. The new object recognition experiment consisted of three stages. The mice were placed individually in the new object recognition response box (bottom side length: 40 × 40 × 40 cm) and explored freely for 5 minutes. After the exploration, the mice were removed. Subsequently, two identical cylinders (Cylinder 1 and Cylinder 2) were placed in two opposite areas of the box, and the mouse moved freely in the box again for 5 minutes. Finally, object Cylinder 1 was left unchanged, and object Cylinder 2 was replaced with a square (cube). The mouse was recorded moving around in the box for 5 minutes.

## DNA extraction, shotgun metagenomic sequencing, and data quality control

The mouse fecal samples were thawed, vortexed, and mixed. Subsequently, 300 mg of the fecal sample was added to 1 mL of phosphate buffered saline buffer, shaken, and mixed well. The centrifuge was used to remove impurities at low speed (10 minutes, 4℃, 800 × $g$), and the sample supernatant was pipetted. Bacterial enrichment was achieved by centrifuging the samples at high speed to collect the organisms (10 minutes, 4℃, 13,000 × $g$). DNA quality was detected using the QIAamp Fast DNA fecal kit (Qiagen, Hillden, Germany), agarose gel electrophoresis, and a nanodrop spectrophotometer (260 nm/280 nm). Metagenomic sequencing was completed by adopting a high-throughput sequencing system, and in this experiment, the Illumina HiSeq 2500 instrument was used for sequencing. After the sequencing was accomplished, gene library construction was performed, and DNA fragments of about 300 bp in length were selected to construct the library. Paired reads were generated using 150 bp in both forward and reverse directions. The quality control pipeline processed the generated metagenomic reads (http://huttenhower.sph.harvard.edu/kneaddata; v0.7.5), using Trimmomatic (61) to filter out low-quality reads less than 60 nt in length. Bowtie2

(v2.3.5.1) eliminated mouse contamination (62). The 9-week-old mouse metagenomic sequence was used to predict the gut metabolome. First, 1 million reads were performed for each sample using seqtk (https://github.com/lh3/seqtk). Secondary sampling reads were compared using DIAMOND. The gene abundance of the sample was then calculated based on the best hit rate of the gene. MelonnPan software is a prediction tool used to convert a table of the relative abundance of predicted metabolite compounds based on gene abundance (63). DIAMOND software is also used to annotate the CAZy family (64).

## Neurotransmitter testing

Fifty-five common neurotransmitters in the blood were tested by liquid chromatography tandem mass spectrometry (LC-MS/MS). Metabolite extraction was performed using a standardized procedure provided by Wuhan MetWare Biotechnology Co., Ltd. (www.met-ware.cn) (65, 66). The entire experimental operation is carried out in a low-temperature environment. Samples stored in the −80℃ refrigerator were thawed and then vortexed for 10 seconds to thoroughly homogenize the samples on ice. Weigh 50 µL of the homogenized sample into 250 µL of methanol solution, shake for 5 minutes, and leave for 5 minutes. Repeat the previous step once and centrifuge the sample at 4℃ (10 minutes at 13,000 $\times$ $g$). Exactly 100 µL of sample supernatant was transferred to the injection vial and analyzed using LC-MS/MS, where ultra-performance liquid chromatography (ExionLC AD) and MS/MS (QTRAP 6500+) formed the data acquisition instrument system.

## Statistical analysis

All data are shown as mean ± standard deviations (SD) or standard error of the mean. All data were analyzed statistically using R software (v.4.1.0), SPSS (IBM SPSS Statistics 25), and GraphPad (GraphPad Software, San Diego, CA, USA). A Wilcoxon rank-sum test with Bonferroni correction or one-way analysis of variance followed by Least Significant Difference post hoc analysis was used for multiple pairwise comparisons. In the graphs, $P < 0.05$ indicates significance and is indicated by *; $P < 0.01$ indicates strong significance and is indicated by **; $P < 0.001$ indicates extreme significance and is indicated by ***. In this study, the heatmaps are drawn by "pheatmap," the vioplot is drawn by "vioplot," and the PCoA analysis is built by the "ade4" package. The "ggplot2" package was used to box plots, the "psych" package was used to calculate the Pearson coefficients, and the "igraph" package was used for correlation plot visualization.

## ACKNOWLEDGMENTS

We are profoundly thankful to the Research Fund for the National Key R&D Program of China (2022YFD2100702), the National Natural Science Foundation of China (U22A20540), the Inner Mongolia Science and Technology Major Projects (2021ZD0014), and the earmarked fund for CARS36.

Z.M. and L.C. contributed equally to this work. H.Z. conceived the concept and designed the study. Z.M. and Y.Z. carried out the experimental manipulations. Z.M. and L.C. contributed to the data analysis of the experiment. J.Z. and L.C. wrote the paper. All the authors agreed to publish the article after discussion.

## AUTHOR AFFILIATIONS

[1]Inner Mongolia Key Laboratory of Dairy Biotechnology and Engineering, Inner Mongolia Agricultural University, Hohhot, Inner Mongolia, China
[2]Key Laboratory of Dairy Products Processing, Ministry of Agriculture and Rural Affairs, Inner Mongolia Agricultural University, Hohhot, Inner Mongolia, China
[3]School of Food Science and Engineering, Hainan University, Haikou, China
[4]School of Chemistry and Biological Engineering, University of Science and Technology Beijing (USTB), Beijing, China

## AUTHOR ORCIDs

Lin Chen http://orcid.org/0009-0000-6298-4672
Jiachao Zhang http://orcid.org/0000-0001-8099-6749
Heping Zhang http://orcid.org/0000-0002-4140-7486

## FUNDING

| Funder | Grant(s) | Author(s) |
|---|---|---|
| Research Fund for the National Key R&D Program of China | 2022YFD2100702 | Heping Zhang |
| MOST \| National Natural Science Foundation of China (NSFC) | U22A20540 | Heping Zhang |
| Inner Mongolia Science and Technology Major Projects | 2021ZD0014 | Heping Zhang |
| CARS36 | | Heping Zhang |

## DATA AVAILABILITY

The sequence data reported in this paper have been deposited in the NCBI database (accession no. PRJNA966790, metagenomic sequencing data).

## ADDITIONAL FILES

The following material is available online.

Open Peer Review

**PEER REVIEW HISTORY (review-history.pdf).** An accounting of the reviewer comments and feedback.

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
