## [Reviewer comments · mSystems]

Bifidobacterium animalis subsp. lactis Probio-M8 alleviates abnormal behavior and regulates gut microbiota in a mouse model suffering autism

Zhuangzhuang Miao, Lin Chen, Yong Zhang, Jiachao Zhang, and Heping Zhang

Corresponding Author(s): Jiachao Zhang, Hainan University

Review Timeline:

Submission Date:

September 20, 2023

Accepted:

November 8, 2023

Editor: Hongwei Zhou

Reviewer(s): The reviewers have opted to remain anonymous.

Transaction Report:

DOI: <https://doi.org/10.1128/msystems.01013-23>

Re: mSystems01013-23 (Bifidobacterium animalis subsp. lactis Probio-M8 alleviates abnormal behavior and regulates gut microbiota in a mouse model suffering autism)

Dear Prof. Jiachao Zhang:

Your manuscript has been accepted, and I am forwarding it to the ASM production staff for publication. Your paper will first be checked to make sure all elements meet the technical requirements. ASM staff will contact you if anything needs to be revised before copyediting and production can begin. Otherwise, you will be notified when your proofs are ready to be viewed.

Featured Image Submissions: If you would like to submit a potential Featured Image, please email a file and a short legend to mSystems@asmusa.org. Please note that we can only consider images that (i) the authors created or own and (ii) have not been previously published. By submitting, you agree that the image can be used under the same terms as the published article. File requirements: square dimensions (4" x 4"), 300 dpi resolution, RGB colorspace, TIF file format.

Sincerely,
Hongwei Zhou
Editor
mSystems